# Modified Industrial Three-Dimensional Polylactic Acid Scaffold Cell Chip Promotes the Proliferation and Differentiation of Human Neural Stem Cells

**DOI:** 10.3390/ijms23042204

**Published:** 2022-02-17

**Authors:** Gyeong-Ji Kim, Kwon-Jai Lee, Jeong-Woo Choi, Jeung Hee An

**Affiliations:** 1Department of Biomedical Engineering, Sogang University, Seoul 04107, Korea; kgj8495@hanmail.net; 2Department of Food and Nutrition, KC University, Seoul 07661, Korea; 3College of H-LAC, Daejeon University, Daejeon 34520, Korea; kjlee@dju.kr; 4Department of Chemical and Biomolecular Engineering, Sogang University, Seoul 04107, Korea

**Keywords:** 3D culture, industrial polylactic acid, cell chip, gold nanoparticles, neural stem cell

## Abstract

In this study, we fabricated a three-dimensional (3D) scaffold using industrial polylactic acid (PLA), which promoted the proliferation and differentiation of human neural stem cells. An industrial PLA 3D scaffold (IPTS) cell chip with a square-shaped pattern was fabricated via computer-aided design and printed using a fused deposition modeling technique. To improve cell adhesion and cell differentiation, we coated the IPTS cell chip with gold nanoparticles (Au-NPs), nerve growth factor (NGF) protein, an NGF peptide fragment, and sonic hedgehog (SHH) protein. The proliferation of F3.Olig2 neural stem cells was increased in the IPTS cell chips coated with Au-NPs and NGF peptide fragments when compared with that of the cells cultured on non-coated IPTS cell chips. Cells cultured on the IPTS-SHH cell chip also showed high expression of motor neuron cell-specific markers, such as HB9 and TUJ-1. Therefore, we suggest that the newly engineered industrial PLA scaffold is an innovative tool for cell proliferation and motor neuron differentiation.

## 1. Introduction

Three-dimensional (3D) bioprinting technologies can be classified into two forms depending on whether or not live cells are printed directly on the construct [1]. Moreover, the fabrication strategies can be divided into extrusion-based, droplet-based, and laser-assisted bioprinting strategies [1,2]. Fused deposition modeling based on extrusion-based systems has been widely applied owing to its advantages of relatively high speed and low cost [1]. The microenvironment of 3D scaffolds promotes cell adhesion, cell growth, cell differentiation, production of extracellular substrates, and cell–cell interactions [3,4]. The printable biomaterials can be divided into hard biomaterials (polymers, ceramics, or metals) and soft materials (hydrogels, comprising synthetic or natural polymers) [1,5]. In particular, polylactic acid (PLA) has recently been used as a scaffold for hard biomaterials because of its improved biocompatibility, excellent bioresorbability, biodegradability, and formation of nontoxic by-products [6]. Some studies using PLA materials have applied the most stringent purification techniques for ensuring the biosafety of tissue engineering and suitability for the microenvironment of mesenchymal stem cells [7,8]. Although the biocompatibility of a scaffold is important for cell growth, biocompatible materials are also expensive [9]. Furthermore, biodegradable PLA possesses characteristic mechanical strength but does not possess the bioactive surface properties required to regulate cellular functions, including extracellular matrix (ECM) secretion and cellular regeneration [10]. Thus, we suggest the application of low-cost industrial PLA materials to offer an optimal environment for regulating cellular functions. Industrial PLA materials are characterized by the emission of particles and volatile organic compounds during printing [11]. The emission of hazardous particles in industrial PLA depends on many factors such as the filament brand, color, extrusion temperature, and feed rate [11,12,13,14,15,16]. Zhang et al. [11] reported that PLA-emitted particles triggered a toxic response in human tumorigenic lung epithelial cells, rat alveolar macrophages, and mouse models. For this reason, industrial PLA is less frequently used as a scaffold material for providing an optimal cellular microenvironment. Thus, we have been focusing on improving the application of industrial PLA as a biomaterial through modification of the surface to increase biocompatibility, cell proliferation, and cell differentiation efficiency. 

Neurodegenerative diseases such as Parkinson’s disease, Huntington’s disease, Alzheimer’s disease, and amyotrophic lateral sclerosis are characterized by the loss or dysfunction of groups of neurons [17]. Recently, several studies have reported the generation of motor neurons in culture from various types of stem cells, including embryonic stem cells, induced pluripotent stem cells and neural stem cells (NSCs) [18]. The natural 3D architecture provides structural support and nutrition, maintains NSCs, and influences subsequent cell function [19]. This effect is adjusted by extracellular and physical cues such as nerve growth factor (NGF), brain-derived neurotrophic factor, morphogens, cell–cell interactions, and cell–ECM interactions [20,21,22]. Furthermore, enhanced transcription of markers of the motor neuron lineage in NSC spheroids compared with that in monolayer cells has been reported [23]. Scaffolds may also influence the differentiation of stem cells into neural lineages for nervous system repair [24]. Bhavika et al. [25] reported that poly-ε-caprolactone fibrous scaffolds promoted the proliferation and differentiation of glial cells. Both HB1.F3.Olig2 (F3.Olig2) and F3 cells have been found to express neuronutrition factors responsible for neuroprotection and regeneration activities [26]. In addition, human NSCs transduced with the oligodendrocyte transcription factor 2 (OLIG2) gene expressed motor neuron-specific genes, including homeobox 9 (*HB9*), after treatment with sonic hedgehog (SHH) protein [27]. However, F3.Olig2 NSCs have not been reported to differentiate into motor neurons in 3D form.

In this study, we developed a low-cost industrial PLA three-dimensional scaffold (IPTS) with reduced cellular toxicity. The IPTS was printed by fused deposition modeling, which resulted in the proliferation, 3D formation, and differentiation of NSCs into motor neurons. Furthermore, to improve the biocompatibility of the industrial PLA material, various materials, including gold nanoparticles (Au-NPs), beta-NGF protein, and an NGF peptide fragment, were coated on the IPTS cell chip (Figure 1). To our knowledge, this is the first report of a modified industrial PLA surface scaffold, which can be easily employed in other studies to mimic the in vivo conditions of motor neuron disease. 

## 2. Results and Discussion

### 2.1. Properties of the IPTS Cell Chip

We first observed the visible shape of the IPTS cell chip through a photograph and optical image analysis. Figure 2A,B shows photographs and optical images of the IPTS cell chip with a pore size of 210 µm. We also measured the physical properties of the IPTS cell chip, including the weight (5.53 ± 0.21 mg), filament diameter (207.57 ± 12.15 µm), and pore size (289.67 ± 21 µm) (Table 1). Printed filaments play an important role in providing a high surface-to-structure ratio with the use of a porous structure that is interconnected and has micropillars to accommodate high-density cells [28]. Additionally, scaffolds with appropriate pore sizes and porosities provide sufficient cell–cell interactions and a microenvironment suitable for cell migration, proliferation, and differentiation [29]. Several studies have reported the need for pore sizes of at least 100 μm in diameter for the successful exchange of oxygen and nutrients for cell growth [30]. Moreover, Jung et al. [31] reported that scaffolds with pore sizes >300 μm can provide a favorable growth environment for cell survival and promote the exchange of nutrients. Thus, our IPTS structure was confirmed to be capable of providing an environment suitable for cell growth.

Young’s modulus and tensile strength were also measured to investigate the mechanical properties of the IPTS cell chip. As shown in Table 1, Young’s modulus and tensile strength of the IPTS cell chip were 29.59 MPa and 13.19 MPa, respectively. PLA is a biodegradable synthetic substance made from dextrose that is extracted from bio-based materials such as cellulose or corn [32]. Therefore, PLA materials are brittle and generally exhibit relatively small deformations at break points [33]. Printed scaffolds are significantly less stiff than solid plastics [34].

We observed the surface characteristics of the IPTS cell chip through atomic force microscopy (AFM) analysis. The AFM image of the IPTS cell chip showed a flat and smooth surface (Figure 2C). PLA hydrophobicity renders bone cell attachment and proliferation difficult [35]. It has been reported that to enhance the surface of an IPTS cell chip, a surface modification method using a material capable of increasing the surface adhesive force, wettability, roughness, and the number of built-in functional groups must be considered [36]. Some studies reported increased cell proliferation after surface roughening [7], surface modification by peptides, hyaluronic acid incubation, addition of collagen, and mussel adhesive proteins [37]. For surface modification of the IPTS cell chip, we examined the effect of coating the surface with Au-NPs, beta-NGF protein, NGF peptide fragments, and SHH protein. 

### 2.2. Properties of IPTS Cell Chips Coated with Au-NPs, Beta-NGF Protein, NGF Peptide Fragments, and SHH Protein

The physical properties of the Au-NP colloidal solution coating on the IPTS cell chip were examined via ultraviolet-visible (UV-vis) spectroscopy. Figure 3A shows the UV-vis wavelength spectra of Au-NPs colloidal solutions recorded from 380 to 780 nm. The corresponding peaks of Au-NPs synthesized at 90 °C were recorded at 519 nm. In addition, according to the published literature, a lower wavelength value for the Au-NPs UV-vis peak is highly correlated with particles of a smaller diameter [37]. We also investigated the surface roughness through AFM analysis. The AFM image showed that the surface of IPTS-Au-NPs is rough with a randomly crude morphology (Figure 3B). Such a rough surface is expected to increase cell adhesion.

Fourier transform infrared spectroscopy (FTIR) was performed to confirm that the various coating materials were well-deposited on the IPTS cell chip. As shown in Figure 3C, FTIR spectra showed the characteristic peaks of the IPTS surface at 866, 1088, 1184, 1454, and 1750 cm^−1^. The FTIR spectra of IPTS coated with Au-NPs showed peaks at 697, 757, 1002, 1060, 1492, and 2961 cm^−1^; the peaks of IPTS-NGF protein were detected at 786, 1074 and 1257 cm^−1^; the peaks of IPTS-peptide fragment NGF were observed at 1084 and 1179 cm^−1^; and the IPTS-SHH protein showed peaks at 997, 1067, 1178, 1492 and 1628 cm^−1^. 

### 2.3. Proliferation of NSCs Cultured on Modified IPTS Cell Chips

The biocompatibility of F3.Olig2 cells cultured on a two-dimensional (2D) plate and on the IPTS coated with Au-NPs, beta-NGF protein, SHH protein, and NGF peptide fragments is shown in Figure 4A. At 3 days, viability was the highest in cells cultured on the 2D plate, followed by the cells cultured on the IPTS-Au-NPs and IPTS-peptide fragment NGF. Additionally, the cell viability in the IPTS-NGF protein group was increased compared with that in the IPTS group. The cytotoxicity of Au-NPs is associated with the size, surface chemistry, shape, concentration of Au-NPs, and type of cells [38]. Pan et al. [39] reported the effects of Au-NPs that were stabilized in triphenylphosphine with diameters ranging from 0.8 to 15 nm on epithelial cells, endothelial cells, macrophages, and fibroblasts cells. The results showed that the larger Au-NPs of 15 nm were nontoxic at concentrations of up to 100-fold relative to the smaller-sized Au-NPs [39]. In addition, a 3D scaffold constructed using an industrial ABS material coated with Au-NPs showed increased safety [40]. In general, NGF disruption is a characteristic of neurotrophies and is related to cell survival, differentiation, neurite outgrowth, and apoptosis [41]. Thus, the surface modified with Au-NPs and NGF protein did not exhibit cytotoxicity. However, IPTS-SHH protein decreased cell viability compared with IPTS. Bermudez et al. [42] reported similar results with A549 (lung adenocarcinoma) cells and H520 (squamous carcinoma) cells treated with recombinant human SHH; no changes in cell number or cell survival were observed at 3 days and 5 days. Overall, these results demonstrated that the IPTS coated with Au-NPs, beta-NGF protein, and NGF peptide fragments decreased the cytotoxicity of particles emitted from the scaffold. 

### 2.4. Comparison of NSC Morphology in Various Modified IPTS Cell Chips

Cell surface topography and contact induction are considered to be key cues in many developmental processes such as neurogenesis [43]. The morphology of NSCs cultured on the IPTS cell chips coated with Au-NPs, beta-NGF protein, NGF peptide fragments, or SHH protein was determined using scanning electron microscopy (SEM), as shown in Figure 4B. The cells cultured on the non-coated IPTS cell chip showed a layered form and filled the pores of the scaffold, whereas the cells cultured on the IPTS-AuNP cell chip attached to the surface of the chip and produced neurites. In addition, cells cultured on the IPTS-beta-NGF protein cell chip formed layers that connected with the neurites, whereas the IPTS-NGF peptide fragment cell chip produced a small amount of neurites in cells that were attached to the pillars of the scaffold. Finally, the morphology of cells cultured on the IPTS-SHH protein cell chip grew in layers on the pillars of the scaffold, and these cells produced many neurites. Generally, traditional 2D cultures promote the differentiation of human neural precursor cells, which form neural networks that appear to mature and stabilize relatively quickly in comparison with those of 3D cultures [43]. However, Spijkers et al. [44] reported that 3D cultures enhanced the ability of cells to differentiate, form intricate networks, and express distinct gene profiles. Thus, our IPTS cell chip can be used as a cell differentiation platform that is more pertinent than a 2D culture system.

### 2.5. Motor Neuron Differentiation of NSCs Cultured on IPTS Cell Chips Coated with Au-NPs, Beta-NGF Protein, or NGF Peptide Fragments

Immunocytochemical and Western blot analyses of F3.Olig2 cells cultured on IPTS cell chips demonstrated that 3D-cultured cells express an early-stage motor neuron marker (HB9) and early neural marker (TUJ-1) [45,46]. HB9 was expressed in aggregated cells on the IPTS cell chip (Figure 5A). Additionally, cells cultured on the IPTS-SHH protein cell chip had increased HB9 expression levels when compared with cells cultured on the non-coated IPTS cell chip. Furthermore, the spheroids formed in the IPTS cell chips coated with beta-NGF protein and NGF peptide fragments expressed HB9. However, HB9 expression in the IPTS-AuNP cell chips was decreased when compared with that of the IPTS cell chip. 

The immunocytochemical images of TUJ-1 expression are shown in Figure 6. TUJ-1 expression was the highest in the IPTS-SHH protein cell chip. In addition, the expression of TUJ-1 was increased in the IPTS-AuNP and IPTS-beta-NGF protein cell chips when compared with that in the IPTS cell chip. However, cells incubated in the IPTS-NGF peptide fragment cell chip had reduced TUJ-1 expression compared with cells cultured on the IPTS cell chip. Western blot analysis showed that the protein levels of TUJ-1 were significantly increased in the IPTS cell chips coated with SHH protein (17.65-fold), Au-NPs (13.83-fold), and beta-NGF protein (3.83-fold) when compared with those in the IPTS cell chip (Figure 6). As cells cultured in the IPTS-SHH protein cell chip promoted both TUJ-1 and HB9 protein expression, it can be suggested that there is enhanced motor neuron differentiation. SHH is widely used to promote the differentiation of human-induced pluripotent stem cells and embryonic stem cells into motor neurons [47]. Lee et al. [27] reported that human NSCs transduced with the OLIG2 gene (F3.Olig2) expressed motor neuron-specific phenotypes, such as HB9 protein expression, following treatment with SHH protein. Au-NPs are also used for the stimulation and modulation of neural activity, including the enhancement of neurite outgrowth, neuron depolarization, modulation of intracellular calcium signaling, and suppression of neuronal activity [48]. TUJ-1 protein expression was increased by 3.83-fold in cells cultured in the IPTS-beta-NGF protein cell chip compared with that in cells cultured in the IPTS cell chip. NGF is a neurotrophic factor critical for the survival and maintenance of sympathetic and sensory neurons [18]. In addition, NGF facilitates the cytoskeletal rearrangements necessary for neurite outgrowth [18]. These results showed that IPTS cell chips coated with SHH protein could enhance motor neuron differentiation.

## 3. Materials and Methods

### 3.1. Fabrication of the IPTS Cell Chip Using a 3D Bioprinter 

The IPTS cell chip was designed to have a two-layer scaffold with a square-shaped pattern (XYZ) (8 mm × 8 mm × 0.2 mm) using the NewCreatorK software program (Version 1.57.70, ROKIT INVIVO Corp.; Seoul, Korea). The IPTS cell chip was then fabricated with a 3D bioprinter (Rokit In vivo, Seoul, Korea). The printing parameters of the chip are listed in Table 2. The PLA white filament of 1.75 mm (3D KNT, Seoul, Korea) was printed through the 200 μm extruder nozzle. The PLA white filament was then inserted into the extruder feed throat and printed by extrusion at 210 °C on a 60 mm Petri dish. The IPTS cell chip with a 40% fill density and line fill pattern was printed with the printing speed set to 7 mm/s. The IPTS cell chip printed on the glass cover slip was covered with the plastic chamber of a Nunc Lab-Tek chamber slide (Thermo Fisher Scientific, Waltham, MA, USA) for cell culture.

### 3.2. Surface Modification of the IPTS Cell Chip to Generate Motor Neurons 

Au-NPs were coated onto the IPTS cell chip to enhance biocompatibility. Once the coverslip with the printed scaffold was placed into a 1 cm^3^ Nunc Lab-Tek chamber slide, it was fixed in place with polydimethylsiloxane. Au-NPs were synthesized using a chemical method modified from Turkevich and Frens [49,50,51]. The process required the reduction of auric ions (Au^3+^) from chloroauric acid trihydrate (HAuCl_4_·3H_2_O) (Kojima Chemicals, Saitama, Japan), which was initiated by the citrate component of trisodium citrate dihydrate salt (C_6_H_5_Na_3_O_7_·2H_2_O) (Samchun Pure Chemicals, Pyeongtaek, Korea). Briefly, a 500 µL solution containing 0.25 mM of Au^3+^ and 1.5 mM of citrate dissolved in deionized water was added to each chamber slide. Thereafter, the mixture was stirred at 50 rpm on an orbital shaker for 5 min, and the scaffold was incubated in the chamber slide with the solution in a preheated dry oven at 90 °C for 60 min (SW-DO002, Seoul, Korea). Finally, the scaffolds were cooled to 36 °C in an oven and then extracted. Synthesized Au-NPs coated the surface of the IPTS cell chip (IPTS-Au-NPs cell chip) by deposition. The UV-vis transmittance spectra of the Au-NPs were recorded in the range of 380–780 nm using a Shimadzu UV mini-1240 spectrophotometer (Kyoto, Japan).

An aqueous solution containing 0.1 M 1-ethyl-3-(3-dimethylaminopropyl) carbodiimide (EDC) and 0.1 M N-hydroxysuccinimide (NHS) was added to the IPTS cell chip for 2 h. The chip was then immersed in 500 µL of 100 ng/mL human beta-NGF recombinant protein solution (Thermo Fisher Scientific, Waltham, MA, USA) or 100 ng/mL SHH protein and incubated for 24 h at 4 °C. The IPTS cell chip was washed with phosphate-buffered saline (PBS) thrice for 5 min and dried on a clean bench for 4 h. The IPTS-beta-NGF and IPTS-SHH protein cell chips were stored at 4 °C. In addition, IPTS cell chips that were activated by EDC and NHS were coated with an NGF peptide fragment, which was synthesized using the sequence IQAEPHSESNVPAGHC (Peptron, Deajeun, Korea). In short, 500 µL of 1 µg/mL NGF peptide fragment solution was added to the IPTS cell chip and incubated for 24 h at 4 °C. The surface of the IPTS cell chip was rinsed with PBS once for 5 min, dried on a clean bench, and stored at 4 °C. 

### 3.3. Characterization of the Surface of Modified IPTS Cell Chips

The filament diameter and pore size of the IPTS cell chip were measured using optical images. The Young’s modulus and tensile strength of the chip were measured using a texture analyzer (TAXT plus/50 Stable Microsystems, Godalming, UK). The surfaces of the IPTS and IPTS-Au-NPs cell chips were observed using AFM (XE-100; Park system Inc., Suwon, Korea) equipped with a non-contact cantilever. The AFM equipment was operated in tapping mode equipped with NCHR, which has a resonance frequency of 320 kHz. The scanning size of the IPTS surface was 1 μm^3^, and a scan rate of 1 Hz was used. The AFM images were edited using the XEI image processing program (Park System Inc.). The IPTS surface coated with Au-NPs, beta-NGF protein, SHH protein, or IPTS-NGF peptide fragment was measured using FTIR in the spectral range of 650–4000 cm^–1^ (Cary 630 FTIR spectrometer, Agilent Technologies, Santa Clara, CA, USA). IR spectra were collected using the Agilent MicroLab PC software (Agilent Technologies, Inc., Foster City, CA, USA).

### 3.4. Culturing Cells on IPTS Cell Chips with Modified Surfaces

The F3.Olig2 NSC line that overexpresses OLIG2 was generated by transfection of F3 cells with the retroviral vector pLPCX-Olig2, and successfully transfected cells were identified by puromycin resistance. F3.Olig2 NSCs were cultured in Dulbecco’s modified Eagle medium (Welgene, Gyeongsan-si, Korea) containing 10% fetal bovine serum (Gibco, NY, USA) and 1% penicillin (Welgene, Gyeongsan-si, Korea) in a humidified incubator with 5% CO_2_ at 37 °C. The F3.Olig2 NSCs were seeded at a density of 1 × 10^4^ cells/mL in IPTS cell chips coated with Au-NPs, beta-NGF protein, SHH protein, or NGF peptide fragment. The cells were incubated for 7 days to allow them to differentiate into motor neurons.

### 3.5. Cell Proliferation on Surface-Modified IPTS Cell Chips 

The WST-1 assay was performed for assessing cell viability to determine the cytotoxicity of the IPTS cell chip and Au-NPs-coated surfaces. The viability of F3.Olig2 NSCs was determined using the EZ-Cytox cell viability assay kit (DoGenBio Co., Ltd., Seoul, Korea), which is based on the cleavage of tetrazolium salt to water-soluble formazan by succinate-tetrazolium reductase. F3.Olig2 NSCs were cultured in the various modified IPTS cell chips for 3 days. Subsequently, 10% EZ-Cytox reagent was added to the cell culture medium, and the chips were incubated for 30 min in a CO_2_ incubator. Next, the mixture of cell culture medium and EZ-Cytox reagent was added to the 96-well plate at 100 μL (in triplicate). The absorbance was measured using a microplate reader (Asys UVM 340; Biochrom Ltd.; Cambridge, UK) at 460 nm with a reference wavelength of 650 nm. A blank comprising medium without cells was used as a baseline reading. Cell viability was calculated by dividing the average absorbance of cells cultured on the various modified IPTS cell chips by the average absorbance of the cells cultured on a non-modified chip.

### 3.6. Morphology of Cells Cultured on Surface-Modified IPTS Cell Chips

The surface morphologies and metabolic status of F3.Olig2 NSCs grown on the various modified IPTS cell chips were analyzed by SEM (Zeiss MERLIN, Oberkochen, Germany). F3.Olig2 NSCs were cultured for 21 days. 

### 3.7. Immunofluorescence Staining of Differentiated Cells Cultured on Surface-Modified IPTS Cell Chips

F3.Olig2 NSCs cultured on the various modified IPTS cell chips were incubated with TUJ-1 (1:1000) and HB9 (1:1000) primary antibodies overnight and then with goat anti-rabbit Alexa Fluor 488-conjugated antibody (1:200) for 2 h (all from Abcam, Cambridge, UK). The cells were imaged using a Zeiss LSM 710 confocal laser scanning fluorescence microscope (Carl Zeiss MicroImaging, Oberkochen, Germany) with a 488 nm excitation filter and analyzed with LSM imaging software (Carl Zeiss, Jena, Germany). 

### 3.8. Western Blot Analysis of Differentiated Cells Cultured on Surface-Modified IPTS Cell Chips

The cells were homogenized in RIPA lysis buffer containing protease inhibitors (Roche, Mannheim, Germany) and incubated on ice for 30 min, followed by centrifugation at 13,000 rpm for 30 min at 4 °C. Proteins were subjected to sodium dodecyl sulfate-polyacrylamide gel electrophoresis and transferred onto Immobilon-P transfer membranes. Total protein levels were determined using a Bio-Rad protein assay kit. Membranes were blocked with 5% bovine serum albumin prior to incubation with primary antibodies against beta-actin (Cell Signaling Technology, Danvers, MA, USA) or TUJ-1 (R&D Systems, Minneapolis, MN, USA). Membranes were then incubated with either goat anti-rabbit IgG H&L or goat anti-mouse IgG H&L (Abcam, Cambridge, UK) secondary antibody. The antigen–antibody complexes were visualized using enhanced chemiluminescence. Densitometric analysis was performed using C-DiGit Blot Scanner (Li-COR Biosciences, Lincoln, NE, USA).

### 3.9. Statistical Analysis

Statistical analyses were performed with SPSS version 18.0 (SPSS Inc., Chicago, IL, USA). Comparisons between experimental groups were made using a one-way analysis of variance with Duncan’s multiple range post hoc test. Statistical significance was considered at *p* < 0.05. The results are presented as the mean ± standard deviation. 

## 4. Conclusions

This study is the first to demonstrate that an IPTS cell chip can be safely used for cell proliferation and motor neuron differentiation. The IPTS cell chip was coated with Au-NPs, beta-NGF protein, NGF peptide fragment, and SHH protein to enhance cell proliferation and adhesion. The surface of the IPTS cell chip coated with Au-NPs, beta-NGF protein, NGF peptide fragment, and SHH protein was confirmed by FTIR spectra. Cell proliferation increased in the IPTS-Au-NPs cell chip and the IPTS-NGF peptide fragment cell chip when compared with that in cells cultured in the non-coated IPTS cell chip. In addition, the NSCs cultured on the IPTS-SHH protein cell chip produced the most neurites among cell groups on the various surface-modified or non-coated IPTS cell chips. Immunocytochemistry further showed that the protein expression levels of HB9 (motor neuron marker) and TUJ-1 (neural marker) increased significantly in the IPTS-SHH protein cell chip compared with those in cells on IPTS cell chips coated with other materials. Furthermore, Western blotting showed that TUJ-1 protein had the highest expression in NSCs cultured on the IPTS-SHH protein cell chip. These results suggest that a cost-effective IPTS cell chip can be used as a motor neuron differentiation platform. 

## Figures and Tables

**Figure 1 ijms-23-02204-f001:**
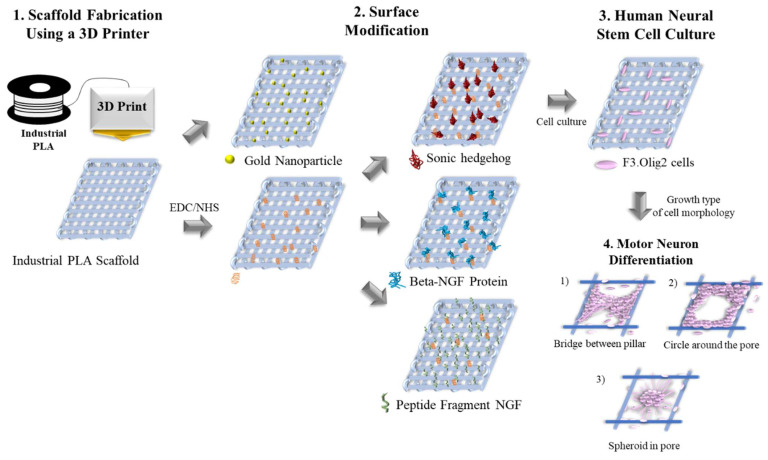
Schematic diagram showing surface modifications of the industrial polylactic acid (PLA) three-dimensional scaffold (IPTS) for motor neuron differentiation.

**Figure 2 ijms-23-02204-f002:**
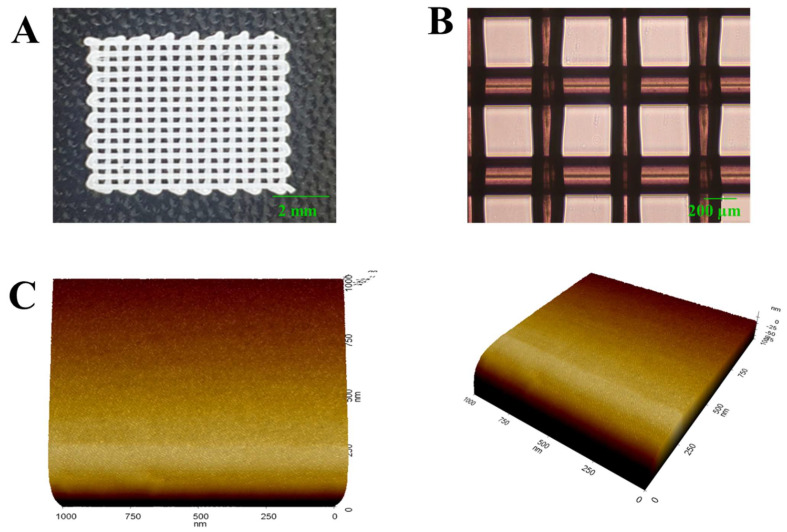
Physical characteristics of the IPTS with 40% infill density, as shown by (**A**) a photograph, (**B**) optical image (scale bar: 200 µm), and (**C**) atomic force microscope image (1 µm^3^) demonstrating the surface roughness of the IPTS.

**Figure 3 ijms-23-02204-f003:**
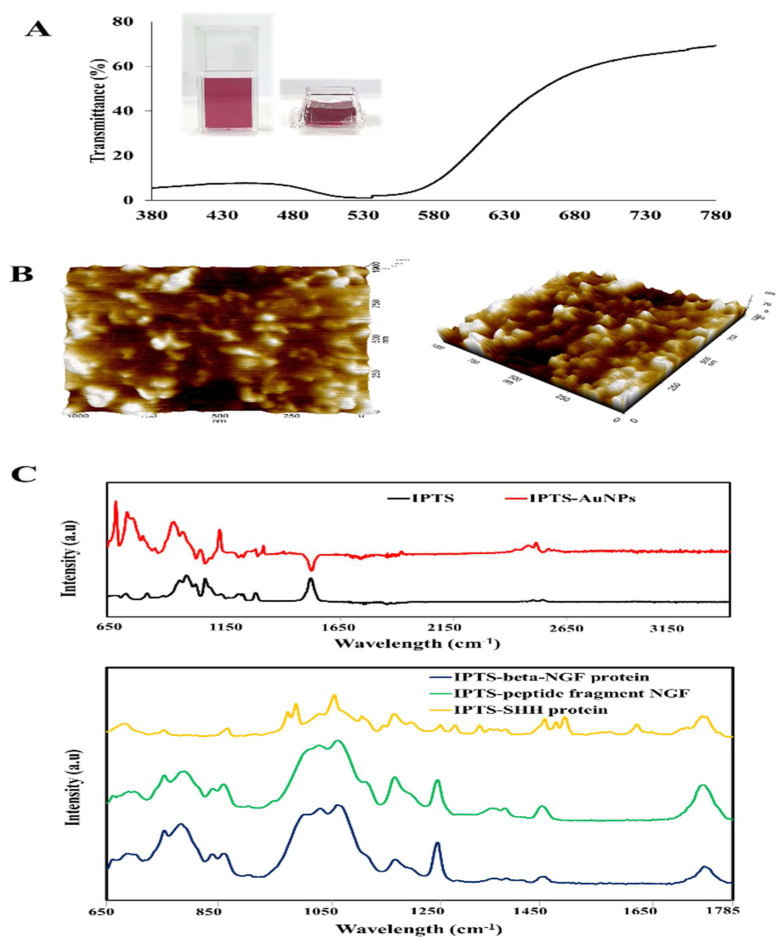
Characteristics of a printed industrial polylactic acid (PLA) three-dimensional scaffold (IPTS) coated with gold nanoparticles (Au-NPs), nerve growth factor (NGF) protein, NGF peptide fragments, or sonic hedgehog (SHH) protein. (**A**) Ultraviolet-visible spectroscopy of Au-NPs in solution. (**B**) Atomic force microscope image (1 µm^3^) showing the surface roughness of IPTS-Au-NPs. (**C**) Fourier transform infrared spectroscopy of non-coated IPTS and IPTS coated with Au-NPs, beta-NGF protein, NGF peptide fragments, and SHH protein.

**Figure 4 ijms-23-02204-f004:**
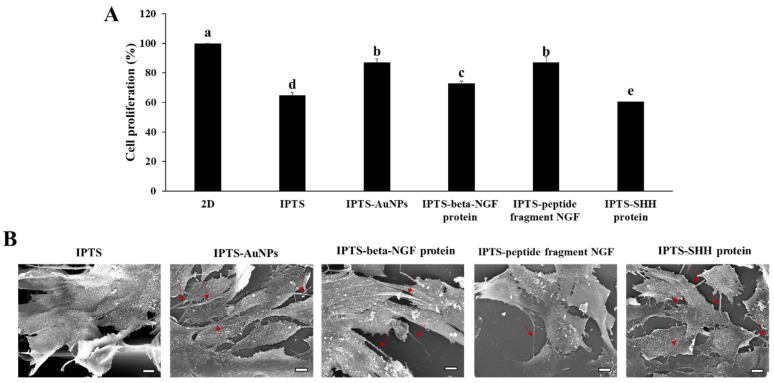
Proliferation of F3.Olig2 NSCs cultured on industrial PLA three-dimensional scaffold (IPTS) chips coated with gold nanoparticles (Au-NPs), nerve growth factor (NGF) protein, NGF peptide fragments, or sonic hedgehog (SHH) protein, compared with that of control cells cultured on a non-coated IPTS or a two-dimensional cell plate. Cells were seeded at a density of 5 × 10^3^ cells/mL. (**A**) Viability of non-differentiated F3.Olig2 NSCs after 2 days of culture using the WST-1 proliferation assay kit. Data are presented as the mean ± standard deviation. The bars not sharing a common lowercase letter (a–e) are significantly different according to Duncan’s multiple range test (*p* < 0.05). (**B**) Scanning electron microscopy images of F3.Olig2 NSCs cultured on the various IPTS chips (scale bar: 10 µm).

**Figure 5 ijms-23-02204-f005:**
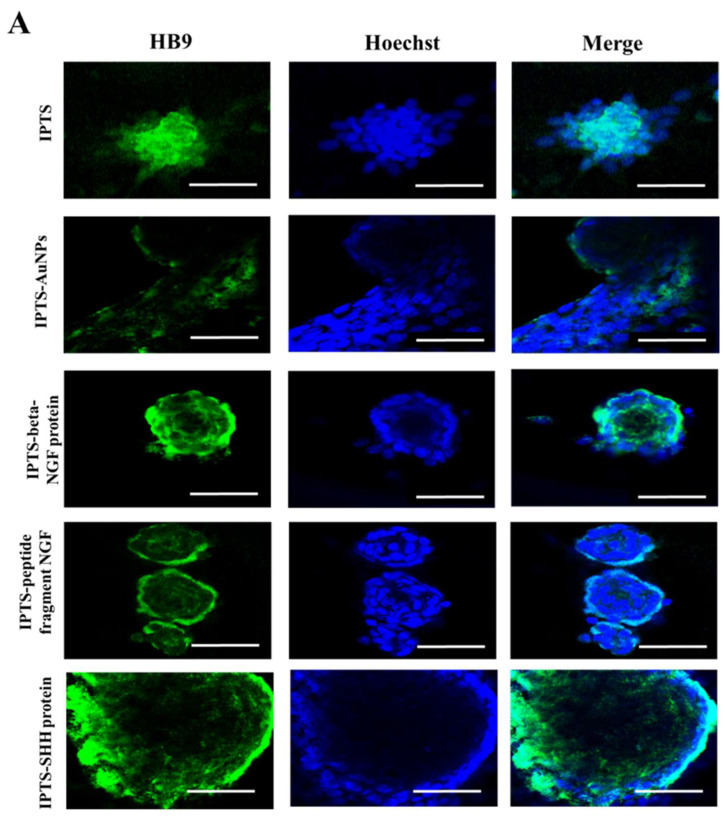
Confocal microscopy images of differentiated F3.Olig2 NSCs cultured on industrial PLA three-dimension scaffold (IPTS) chips coated with gold nanoparticles (Au-NPs), nerve growth factor (NGF) protein, NGF peptide fragments, or sonic hedgehog (SHH) protein. Samples were stained for (**A**) HB9 (green) and (**B**) TUJ-1 (green) using a goat anti-rabbit Alexa Fluor 488 conjugated antibody. Nuclei were stained with Hoechst (blue) (scale bar: 50 µm). The images were obtained using a Carl Zeiss LSM 800 confocal laser scanning microscope and imaged using LSM imaging software (Carl Zeiss).

**Figure 6 ijms-23-02204-f006:**
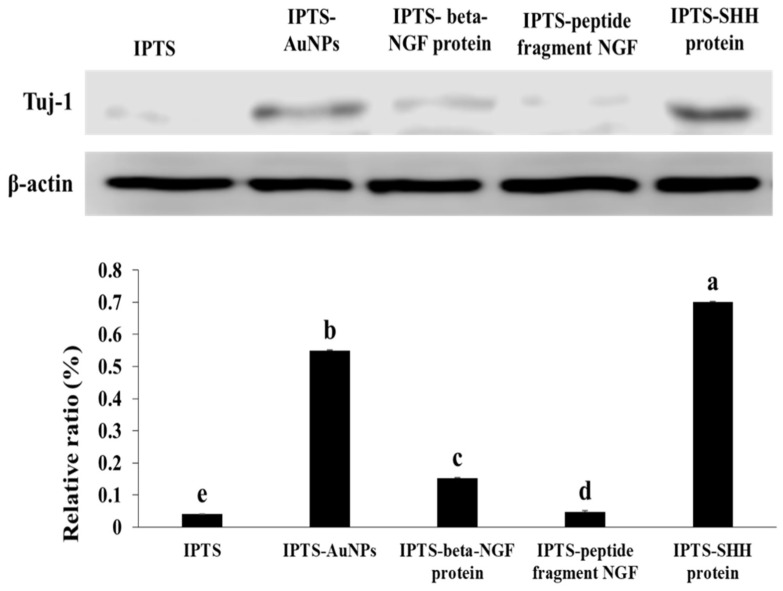
Levels of TUJ-1 protein determined by Western blot. The relative expression level was quantified using ImageJ and normalized to β-actin. Data are presented as the mean ± standard deviation. The bars not sharing a common lowercase letter (a–e) are significantly different according to Duncan’s multiple range test (*p* < 0.05).

**Table 1 ijms-23-02204-t001:** The average weight, filament diameter, pore size, Young’s modulus, and tensile stress of the industrial polylactic acid (PLA) three-dimensional scaffold (*n* = 3).

Weight (mg)	Filament Diameter (µm)	Pore Size (µm)	Young’s Modulus (MPa)	Tensile Strength
5.53 ± 0.21	207.57 ± 12.15	289.67 ± 21.00	29.59 ± 7.58	13.19 ± 2.93

**Table 2 ijms-23-02204-t002:** Printer settings for producing the industrial polylactic acid (PLA) three-dimensional scaffold.

Nozzle Diameter	0.2 mm
Layer height	0.1 mm
Fill density	40%
Fill pattern	Lines
Extruder temperature	210 °C
Bed temperature	25 °C
Speed of print moves	7 mm/s

## Data Availability

Not applicable.

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
