# Peer review of "Modified Industrial Three-Dimensional Polylactic Acid Scaffold Cell Chip Promotes the Proliferation and Differentiation of Human Neural Stem Cells"

_ijms, 2022, doi:10.3390/ijms23042204_

Round 1

Reviewer 1 Report

  1. Differences between industrial PLA and the one used for scaffold printing would be a good addition to indicate the choice of material
  2. Methods section need to be be improved -  details of design and printing processes should be added

Author Response

Responses to Reviewers’ Comments

Manuscript ID: ijms-1585984

Title: Modified industrial 3D polylactic acid scaffold cell chip promotes proliferation and differentiation of human neural stem cells

Reviewer 1

  • Differences between industrial PLA and the one used for scaffold printing would be a good addition to indicate the choice of material.

à Thank you for your valuable comments. We added the following text in the Introduction to explain the differences between industrial PLA and that used for scaffold printing in this study:

Page 1-2:

  1. Introduction

Three-dimensional (3D) bioprinting technologies can be classified into two forms depending on whether or not live cells are printed directly on the construct [1]. Moreover, the fabrication strategies can be divided into extrusion-based, droplet-based, and laser-assisted bioprinting strategies [1,2]. Fused deposition modeling based on extrusion-based systems has been widely applied owing to its advantages of relatively high speed and low cost [1]. The microenvironment of 3D scaffolds promotes cell adhesion, cell growth, cell differentiation, production of extracellular substrates, and cell-to-cell interactions [3,4]. The printable biomaterials can be divided into hard biomaterials (polymers, ceramics, or metals) and soft materials (hydrogels, comprising synthetic or natural polymers) [1,5]. In particular, polylactic acid (PLA) has recently been used as a scaffold for hard biomaterials because of its improved biocompatibility, excellent bioresorbability, biodegradability, and formation of nontoxic by-products [6]. Some studies using PLA materials have applied the most stringent purification techniques for ensuring the biosafety of tissue engineering and suitability for the microenvironment of mesenchymal stem cells [7,8]. Although the biocompatibility of a scaffold is important for cell growth, biocompatible materials are also expensive [9]. Furthermore, biodegradable PLA possesses characteristic mechanical strength but does not possess the bioactive surface properties required to regulate cellular functions, including extracellular matrix (ECM) secretion and cellular regeneration [10]. Thus, we suggest the application of low-cost industrial PLA materials to offer an optimal environment for regulating cellular functions. Industrial PLA materials are characterized by the emission of particles and volatile organic compounds during printing [11]. The emission of hazardous particles in industrial PLA depends on many factors such as the filament brand, color, extrusion temperature, and feed rate [11–16]. Zhang et al. [11] reported that PLA-emitted particles triggered a toxic response in human tumorigenic lung epithelial cells, rat alveolar macrophages, and mouse models. For this reason, industrial PLA is less frequently used as a scaffold material for providing an optimal cellular microenvironment. Thus, we have been focusing on improving the application of industrial PLA as a biomaterial through modification of the surface to increase the biocompatibility, cell proliferation, and cell differentiation efficiency.

Neurodegenerative diseases such as Parkinson’s disease, Huntington’s disease, Alzheimer’s disease, and amyotrophic lateral sclerosis are characterized by the loss or dysfunction of groups of neurons [17]. Recently, several studies have reported the generation of motor neurons in culture from various types of stem cells, including embryonic stem cells, induced pluripotent stem cells, and neural stem cells (NSCs) [18]. The natural 3D architecture provides structural support and nutrition, maintains NSCs, and influences subsequent cell function [19]. This effect is adjusted by extracellular and physical cues such as nerve growth factor (NGF), brain-derived neurotrophic factor, morphogens, cell-cell interactions, and cell-ECM interactions [20–22]. Furthermore, enhanced transcription of markers of the motor neuron lineage in NSC spheroids compared with that in monolayer cells has been reported [23]. Scaffolds may also influence the differentiation of stem cells into neural lineages for nervous system repair [24]. Bhavika et al. [25] reported that poly-ε-caprolactone fibrous scaffolds promoted the proliferation and differentiation of glial cells. Both HB1.F3.Olig2 (F3.Olig2) and F3 cells have been found to express neuronutrition factors responsible for neuroprotection and regeneration activities [26]. In addition, human NSCs transduced with the oligodendrocyte transcription factor 2 (OLIG2) gene expressed motor neuron-specific genes, including homeobox 9 (HB9), after treatment with sonic hedgehog (SHH) protein [27]. However, F3.Olig2 NSCs have not been reported to differentiate into motor neurons in 3D form.

  • Methods section need to be be improved details of design and printing processes should be added

à  We added the details of the design and printing processes in the Methods as follows:

    Page 9-10

  1. Materials and Methods

3.1. Fabrication of the IPTS cell chip using a 3D bioprinter

The IPTS cell chip was designed to have a two-layer scaffold with a square-shape pattern (XYZ) (8 mm × 8 mm × 0.2 mm) using the NewCreatorK software program (Version 1.57.70, ROKIT INVIVO Corp.; Seoul, Korea). The IPTS cell chip was then fabricated with a 3D bioprinter (Rokit In vivo, Seoul, Korea). The printing parameters of the chip are listed in Table 2. The PLA white filament of 1.75 mm (3D KNT, Seoul, Korea) was printed through the 200 μm extruder nozzle. The PLA white filament was then inserted into the extruder feed throat and printed by extrusion at 210°C on a 60-mm Petri dish. The IPTS cell chip with a 40% fill density and line fill pattern was printed with the printing speed set to 7 mm/s. The IPTS cell chip printed on the glass cover slip was covered with the plastic chamber of a Nunc Lab-Tek chamber slide (Thermo Fisher Scientific, Waltham, MA, USA) for cell culture.

 We have carefully checked and rectified the formatting and grammatical errors in the manuscript

Reviewer 2 Report

In this manuscript, the author reports the “Modified industrial 3D polylactic acid scaffold cell chip promotes proliferation and differentiation of human neural stem cells”

The authors should address the following questions before getting a possible publication.

Recommendation: Major revision needed as noted.

  1. The authors may discuss if brief about the novelty of the present work
  2. The formatting and grammatical errors in the article need to be checked carefully. There are a lot of grammatical tense errors in the manuscript.
  3. Which excitation filter did the author use for the confocal microscopy images?
  4. What does the error bars stand for in the Fig.4a and Fig.6? That should be mentioned in the Figure captions.
  5. The author should write purpose for each test in one/two sentences (in brief) before explaining the results of the characterization techniques. Therefore, the logic and organization of this part will be enhanced.
  6. The conclusion should be rewritten with important results of the work.
  7. The authors cited some of the relevant research works that have been conducted in this area however there are a few that needs to be included (shown below) in the Introduction section: Advanced healthcare materials, 6(1), 1601118; Macromolecular bioscience, 19(2), 1800236; ACS Applied Materials & Interfaces, 12(46), 51940-51951

Author Response

Responses to Reviewers’ Comments

Manuscript ID: ijms-1585984

Title: Modified industrial 3D polylactic acid scaffold cell chip promotes proliferation and differentiation of human neural stem cells

Reviewer 2

In this manuscript, the author reports the “Modified industrial 3D polylactic acid scaffold cell chip promotes proliferation and differentiation of human neural stem cells”

The authors should address the following questions before getting a possible publication.

Recommendation: Major revision needed as noted.

The authors may discuss if brief about the novelty of the present work

  • The formatting and grammatical errors in the article need to be checked carefully. There are a lot of grammatical tense errors in the manuscript.

à Thank you for your valuable comments. We have carefully checked and rectified the formatting and grammatical errors in the manuscript.

  • Which excitation filter did the author use for the confocal microscopy images?

à We have added the excitation filter information for confocal microscopy images in the Methods as follows:

Page 12

3.7. Immunofluorescence staining of differentiated cells cultured on surface-modified IPTS cell chips

F3.Olig2 NSCs cultured on the various modified IPTS cell chips were incubated with TUJ-1 (1:1000) and HB9 (1:1000) primary antibodies overnight and then with goat anti-rabbit Alexa Fluor 488-conjugated antibody (1:200) for 2 h (all from Abcam, Cambridge, UK). The cells were imaged using a Zeiss LSM 710 confocal laser scanning fluorescence microscope (Carl Zeiss Micro Imaging, Oberkochen, Germany) with a 488-nm excitation filter and analyzed with LSM imaging software (Carl Zeiss, Jena, Germany).

  • What does the error bars stand for in the Fig.4a and Fig.6? That should be mentioned in the Figure captions.

à We have revised the figure captions to clarify the meaning of the error bars, as follows:

   Page 6 and 9

Figure 4. Proliferation of F3.Olig2 NSCs cultured on industrial PLA three-dimensional scaffold (IPTS) chips coated with gold nanoparticles (Au-NPs), nerve growth factor (NGF) protein, NGF peptide fragments, or sonic hedgehog (SHH) protein, compared with that of control cells cultured on a non-coated IPTS or a two-dimensional cell plate. Cells were seeded at a density of 5 × 103 cells/mL. (A) Viability of non-differentiated F3.Olig2 NSCs after 2 days of culture using the WST-1 proliferation assay kit. Data are presented as the mean ± standard deviation. The bars not sharing a common lowercase letter (a–e) are significantly different according to Duncan’s multiple range test (p < 0.05). (B) Scanning electron microscopy images of F3.Olig2 NSCs cultured on the various IPTS chips (scale bar: 10 µm).

Figure 6. Levels of TUJ-1 protein determined by western blot. The relative expression level was quantified using ImageJ and normalized to β-actin. Data are presented as the mean ± standard deviation. The bars not sharing a common lowercase letter (a–e) are significantly different according to Duncan’s multiple range test (p < 0.05).

  • The author should write purpose for each test in one/two sentences (in brief) before explaining the results of the characterization techniques. Therefore, the logic and organization of this part will be enhanced.

à We have added sentences related to the purpose/aims of each experiment before presenting the results of characterization techniques, as follows:

Pages 4–5

  1. Results and Discussion

2.1. Properties of the IPTS cell chip

We first observed the visible shape of the IPTS cell chip through photograph and optical image analysis. Figure 2A and 2B show photographs and optical images of the IPTS cell chip with a pore size of 210 µm. We also measured the physical properties of the IPTS cell chip, including the weight (5.53 ± 0.21 mg), filament diameter (207.57 ± 12.15 µm), and pore size (289.67 ± 21 µm) (Table 1). Printed filaments play an important role in providing a high surface-to-structure ratio with the use of a porous structure that is interconnected and has micropillars to accommodate high-density cells [28]. Additionally, scaffolds with appropriate pore sizes and porosities provide sufficient cell-cell interactions and a microenvironment suitable for cell migration, proliferation, and differentiation [29]. Several studies have reported the need for pore sizes of at least 100 μm in diameter for the successful exchange of oxygen and nutrients for cell growth [30]. Moreover, Jung et al. [31] reported that scaffolds with pore sizes >300 μm can provide a favorable growth environment for cell survival and promote the exchange of nutrients. Thus, our IPTS structure was confirmed to be capable of providing an environment suitable for cell growth.

Young’s modulus and tensile strength were also measured to investigate the mechanical properties of the IPTS cell chip. As shown in Table 1, the Young’s modulus and tensile strength of the IPTS cell chip were 29.59 MPa and 13.19 MPa, respectively. PLA is a biodegradable synthetic substance made from dextrose that is extracted from bio-based materials such as cellulose or corn [32]. Therefore, PLA materials are brittle and generally exhibit relatively small deformations at break points [33]. Printed scaffolds are significantly less stiff than solid plastics [34].

We observed the surface characteristics of the IPTS cell chip through atomic force microscopy (AFM) analysis. The AFM image of the IPTS cell chip showed a flat and smooth surface (Figure 2C). PLA hydrophobicity renders bone cell attachment and proliferation difficult [35]. It has been reported that to enhance the surface of an IPTS cell chip, a surface modification method using a material capable of increasing the surface adhesive force, wettability, roughness, and the number of built-in functional groups must be considered [36]. Some studies reported increased cell proliferation after surface roughening [7], surface modification by peptides, hyaluronic acid incubation, addition of collagen, and mussel adhesive proteins [37]. For surface modification of the IPTS cell chip, we examined the effect of coating the surface with Au-NPs, beta-NGF protein, NGF peptide fragments, and SHH protein.

2.2. Properties of IPTS cell chips coated with Au-NPs, beta-NGF protein, NGF peptide fragments, and SHH protein

 The physical properties of the Au-NP colloidal solution coating on the IPTS cell chip were examined via ultraviolet-visible (UV-vis) spectroscopy. Figure 3A shows the UV-vis wavelength spectra of Au-NPs colloidal solutions recorded from 380 to 780 nm. The corresponding peaks of Au-NPs synthesized at 90°C were recorded at 519 nm. In addition, according to the published literature, a lower wavelength value for the Au-NPs UV-vis peak is highly correlated with particles of a smaller diameter [37]. We also investigated the surface roughness thorough AFM analysis. The AFM image showed that the surface of IPTS-Au-NPs is rough with a randomly crude morphology (Figure 3B). Such a rough surface is expected to increase cell adhesion.

Fourier-transform infrared spectroscopy (FTIR) was performed to confirm that the various coating materials were well-deposited on the IPTS cell chip. As shown in Figure 3C, FTIR spectra showed the characteristic peaks of the IPTS surface at 866, 1088, 1184, 1454, at 1750 cm−1. The FTIR spectra of IPTS coated with Au-NPs showed peaks at 697, 757, 1002, 1060, and 1492 and 2961 cm−1; the peaks of IPTS-NGF protein were detected at 786, 1074, 1257, and 2962 cm−1; the peaks of IPTS-peptide fragment NGF were observed at 1084 and 1179 cm−1; and the IPTS-SHH protein showed peaks at 997, 1067, 1178, 1492, 3.23, and 3249 cm−1.

  • The conclusion should be rewritten with important results of the work.

à We have revised the Conclusion as follows:

   Page 12

  1. Conclusions

This study is the first to demonstrate that an IPTS cell chip can be safely used for cell proliferation and motor neuron differentiation. The IPTS cell chip was coated with Au-NPs, beta-NGF protein, NGF peptide fragment, and SHH protein to enhance cell proliferation and adhesion. The surface of the IPTS cell chip coated with Au-NPs, beta-NGF protein, NGF peptide fragment, and SHH protein was confirmed by FTIR spectra. Cell proliferation increased in the IPTS-Au-NPs cell chip and the IPTS-NGF peptide fragment cell chip when compared with that in cells cultured in the non-coated IPTS cell chip. In addition, the NSCs cultured on the IPTS-SHH protein cell chip produced the most neurites among cell groups on the various surface-modified or non-coated IPTS cell chips. Immunocytochemistry further showed that the protein expression levels of HB9 (motor neuron marker) and TUJ-1 (neural marker) increased significantly in the IPTS-SHH protein cell chip compared with those in cells on IPTS cell chips coated with other materials. Furthermore, western blotting showed that TUJ-1 protein had the highest expression in NSCs cultured on the IPTS-SHH protein cell chip. These results suggest that a cost-effective IPTS cell chip can be used as a motor neuron differentiation platform

  • The authors cited some of the relevant research works that have been conducted in this area however there are a few that needs to be included (shown below) in the Introduction section: Advanced healthcare materials, 6(1), 1601118; Macromolecular bioscience, 19(2), 1800236; ACS Applied Materials & Interfaces, 12(46), 51940-51951

à We added the three suggested references to the Introduction and renumbered the references and citations accordingly:

Page 1-2

  1. Introduction

Three-dimensional (3D) bioprinting technologies can be classified into two forms depending on whether or not live cells are printed directly on the construct [1]. Moreover, the fabrication strategies can be divided into extrusion-based, droplet-based, and laser-assisted bioprinting strategies [1,2]. Fused deposition modeling based on extrusion-based systems has been widely applied owing to its advantages of relatively high speed and low cost [1]. The microenvironment of 3D scaffolds promotes cell adhesion, cell growth, cell differentiation, production of extracellular substrates, and cell-to-cell interactions [3,4]. The printable biomaterials can be divided into hard biomaterials (polymers, ceramics, or metals) and soft materials (hydrogels, comprising synthetic or natural polymers) [1,5]. In particular, polylactic acid (PLA) has recently been used as a scaffold for hard biomaterials because of its improved biocompatibility, excellent bioresorbability, biodegradability, and formation of nontoxic by-products [6]. Some studies using PLA materials have applied the most stringent purification techniques for ensuring the biosafety of tissue engineering and suitability for the microenvironment of mesenchymal stem cells [7,8]. Although the biocompatibility of a scaffold is important for cell growth, biocompatible materials are also expensive [9]. Furthermore, biodegradable PLA possesses characteristic mechanical strength but does not possess the bioactive surface properties required to regulate cellular functions, including extracellular matrix (ECM) secretion and cellular regeneration [10]. Thus, we suggest the application of low-cost industrial PLA materials to offer an optimal environment for regulating cellular functions. Industrial PLA materials are characterized by the emission of particles and volatile organic compounds during printing [11]. The emission of hazardous particles in industrial PLA depends on many factors such as the filament brand, color, extrusion temperature, and feed rate [11–16]. Zhang et al. [11] reported that PLA-emitted particles triggered a toxic response in human tumorigenic lung epithelial cells, rat alveolar macrophages, and mouse models. For this reason, industrial PLA is less frequently used as a scaffold material for providing an optimal cellular microenvironment. Thus, we have been focusing on improving the application of industrial PLA as a biomaterial through modification of the surface to increase the biocompatibility, cell proliferation, and cell differentiation efficiency.

Neurodegenerative diseases such as Parkinson’s disease, Huntington’s disease, Alzheimer’s disease, and amyotrophic lateral sclerosis are characterized by the loss or dysfunction of groups of neurons [17]. Recently, several studies have reported the generation of motor neurons in culture from various types of stem cells, including embryonic stem cells, induced pluripotent stem cells, and neural stem cells (NSCs) [18]. The natural 3D architecture provides structural support and nutrition, maintains NSCs, and influences subsequent cell function [19]. This effect is adjusted by extracellular and physical cues such as nerve growth factor (NGF), brain-derived neurotrophic factor, morphogens, cell-cell interactions, and cell-ECM interactions [20–22]. Furthermore, enhanced transcription of markers of the motor neuron lineage in NSC spheroids compared with that in monolayer cells has been reported [23]. Scaffolds may also influence the differentiation of stem cells into neural lineages for nervous system repair [24]. Bhavika et al. [25] reported that poly-ε-caprolactone fibrous scaffolds promoted the proliferation and differentiation of glial cells. Both HB1.F3.Olig2 (F3.Olig2) and F3 cells have been found to express neuronutrition factors responsible for neuroprotection and regeneration activities [26]. In addition, human NSCs transduced with the oligodendrocyte transcription factor 2 (OLIG2) gene expressed motor neuron-specific genes, including homeobox 9 (HB9), after treatment with sonic hedgehog (SHH) protein [27]. However, F3.Olig2 NSCs have not been reported to differentiate into motor neurons in 3D form.

In this study, we developed a low-cost industrial PLA three-dimensional scaffold (IPTS) with reduced cellular toxicity. The IPTS was printed by fused deposition modeling, which resulted in the proliferation, 3D formation, and differentiation of NSCs into motor neurons. Furthermore, to improve the biocompatibility of the industrial PLA material, various materials, including gold nanoparticles (Au-NPs), beta-NGF protein, and an NGF peptide fragment, were coated on the IPTS cell chip (Figure 1). To our knowledge, this is the first report of a modified industrial PLA surface scaffold, which can be easily employed in other studies to mimic the in vivo conditions of motor neuron disease.

Page 13

References

  1. Cui, H.; Nowicki, M.; Fisher, J.P.; Zhang, L.G. 3D bioprinting for organ regeneration. Adv. Healthc. Mater. 2017, 6, 1601118.
  2. Murphy, S.V.; Atala, A. 3D bioprinting of tissues and organs. Nat. Biotechnol. 2014, 32, 773-785.
  3. Nguyen, A.H.; March, P.; Schmiess-Heine, L.; Burke, P.J.; Lee, A.; Lee, J.; Cao, H. Cardiac tissue engineering: state-of-the-art methods and outlook. J. Biol. Eng. 2019, 13, 57.
  4. Geetha Bai, R.; Muthoosamy, K.; Manickam, S.; Hilal-Alnaqbi, A. Graphene-based 3D scaffolds in tissue engineering: fabrication, applications, and future scope in liver tissue engineering. Int. J. Nanomed. 2019, 2019, 14, 5753-5783.
  5. Ganguly, S.; Das, P.; Itzhaki, E.; Hadad, E.; Gedanken, A.; Margel, S. Microwave-synthesized polysaccharide-derived carbon dots as therapeutic cargoes and toughening agents for elastomeric gels. ACS Appl. Mater. Interfaces 2020, 12, 51940-51951.
  6. Yuan, B.; Zhou, S. Y.; Chen, X. S.; J. Zhejiang. Rapid prototyping technology and its application in bone tissue engineering. Univ. Sci. B 2017, 18, 303-315.
  7. Salerno, A.; Fernández-Gutiérrez, M.; del Barrio, J.S.R.; Domingo, C. Bio-safe fabrication of PLA scaffolds for bone tissue engineering by combining phase separation, porogen leaching and scCO2 drying. J. Supercrit. Fluids 2015, 97, 238-246.
  8. Salerno, A.; Guarino, V.; Oliviero, O.; Ambrosio, L.; Domingo, C. Bio-safe processing of polylactic-co-caprolactone and polylactic acid blends to fabricate fibrous porous scaffolds for in vitro mesenchymal stem cells adhesion and proliferation. Mater. Sci. Eng. C 2016, 63, 512-521.

Round 2

Reviewer 2 Report

The manuscript can be accepted in the present form